

# Identification and characterization of biomarkers associated with endoplasmic reticulum protein processing in cerebral ischemia-reperfusion injury

Liang-da Li[1], Yue Zhou[1] and Shan-fen Shi[2]

[1] Department of Neurology, The People's Hospital Affiliated to Ningbo University, Ningbo, Zhejiang, China
[2] Department of Rheumatology, The People's Hospital Affiliated to Ningbo University, Ningbo, Zhejiang, China

## ABSTRACT

**Background**. Cerebral ischemia (CI), ranking as the second leading global cause of death, is frequently treated by reestablishing blood flow and oxygenation. Paradoxically, this reperfusion can intensify tissue damage, leading to CI-reperfusion injury. This research sought to uncover biomarkers pertaining to protein processing in the endoplasmic reticulum (PER) during CI-reperfusion injury.

**Methods**. We utilized the Gene Expression Omnibus (GEO) dataset GSE163614 to discern differentially expressed genes (DEGs) and single out PER-related DEGs. The functions and pathways of these PER-related DEGs were identified *via* Gene Ontology (GO) and the Kyoto Encyclopedia of Genes and Genomes (KEGG) enrichment analyses. Core genes were pinpointed through protein-protein interaction (PPI) networks. Subsequent to this, genes with diagnostic relevance were distinguished using external validation datasets. A single-sample gene-set enrichment analysis (ssGSEA) was undertaken to pinpoint genes with strong associations to hypoxia and apoptosis, suggesting their potential roles as primary inducers of apoptosis in hypoxic conditions during ischemia-reperfusion injuries.

**Results**. Our study demonstrated that PER-related genes, specifically ADCY5, CAMK2A, PLCB1, NTRK2, and DLG4, were markedly down-regulated in models, exhibiting a robust association with hypoxia and apoptosis.

**Conclusion**. The data indicates that ADCY5, CAMK2A, PLCB1, NTRK2, and DLG4 could be pivotal genes responsible for triggering apoptosis in hypoxic environments during CI-reperfusion injury.

Corresponding author
Liang-da Li, rmliliangda@nbu.edu.cn

## INTRODUCTION

Factors such as external force, thrombosis, and thromboembolic arterial occlusion can lead to cerebral ischemia (CI) (*Bramlett & Dietrich, 2004*), also known as stroke, the second primary cause of death worldwide (*Katan & Luft, 2018*). Prompt restoration of blood circulation is vital for CI treatment. Characterized by the interruption of blood and oxygen

supply, ischemia can induce local tissue hypoxia. When blood flow is restored, reperfusion may exacerbate tissue damage and trigger a severe inflammatory reaction, a phenomenon termed "reperfusion injury" (*Eltzschig & Eckle, 2011*). Ischemia-reperfusion injury can provoke various cellular death events, including apoptosis, necrosis, autophagy-associated cell death, and mitochondrial dysfunction (*Eltzschig & Eckle, 2011*; *Khoshnam, Winlow & Farzaneh, 2017*; *Neag et al., 2022*). Restoration of blood oxygen supply after prolonged ischaemia may further exacerbate brain damage and neurological deficits associated with local inflammation and excess reactive oxygen (ROS) species-induced neuronal cell death (*Jurcau & Simion, 2021*; *Zhang et al., 2022*).

Protein processing in the endoplasmic reticulum (PER) refers to the process where newly synthesized polypeptide chains in eukaryotes initially enter the endoplasmic reticulum (ER), undergo post-translational modifications like glycosylation and disulfide bond formation in the ER cavity, fold and assemble into multi-subunit complexes, and are subsequently transported to the Golgi apparatus (*Kaufman, 2002*; *Malhotra & Kaufman, 2007*). Misfolded proteins bind to BiP and are then degraded by ER-associated degradation (ERAD). Ischemia and hypoxia can cause an accumulation of unfolded proteins in the ER cavity, thus initiating the unfolded protein response (UPR) (*Malhotra & Kaufman, 2007*; *Pan et al., 2021*). The UPR, ERAD, autophagy, and hypoxia signal transduction collectively dictate the extent of ER stress (ERS), which in turn determines the intracellular balance or initiates the death process. The hypoxia-Inducible Factor 1 (HIF-1) Pathway is one of the longest-studied signals (*Semenza, 2007*). HIF-1 induces high concentrations of pro- or anti-apoptotic proteins under different oxygenated conditions, inducing apoptosis at the cellular or local level (*Greijer & Van der Wall, 2004*). Therefore HIF-1 signalling has been extensively studied in recent years as a key target in the treatment of cardiovascular diseases (*Liu et al., 2020*).

In the context of CI-reperfusion, a combination of calcium overload, ROS accumulation, and inflammatory response can trigger ERS. The severity of ERS determines cellular fate: ERS promotes cell survival by initiating UPR in the early stage of CI-reperfusion, whereas severe and persistent ERS and UPR lead to cell apoptosis (*Wang et al., 2022*). Apoptosis remains the predominant mode of cell death in CI-reperfusion (*Zhang et al., 2022*), and interfering with neuronal apoptotic signalling and thus neuroprotection provides a theoretical basis for the clinical prevention and treatment of CI-reperfusion.

It has been reported that inhibiting ERS can reduce the volume of cerebral infarction, suggesting that targeting ERS may be an effective strategy for mitigating brain injury. However, no efficacious targets have yet been identified for effectively alleviating CI-reperfusion injury. Unravelling these complexities may provide assistance in treatment strategies for stroke patients, increasing patient survival and improving the quality of life of stroke survivors. Therefore, this study utilized bioinformatics to analyze public databases of rat models of CI-reperfusion injury, aiming to screen and determine potential therapeutic targets.

## MATERIALS AND METHODS

### Data acquisition and pre-processing

The GSE163614 dataset was retrieved from the National Center for Biotechnology Information (NCBI) database. ENSRNOG identifiers were mapped to gene symbols utilizing the annotation file Rattus_norvegicus.mRatBN7.2.107.gtf. This dataset comprised three CI/reperfusion injured rats and three control rats (*Yi et al., 2021*).

Datasets GSE78731 (*Oh et al., 2018*) and GSE97537 (*Zou et al., 2019*) were also sourced from the Gene Expression Omnibus (GEO) within the NCBI database. Probe identifiers were transformed into gene symbols using the platform-specific file. Entries with a single probe mapping to multiple genes were excluded. When multiple probes were mapped to one gene, their data were averaged. From GSE78731, five Middle Cerebral Artery Occlusion (MCAO) and five control samples were extracted, and from GSE97537, seven MCAO and five sham samples were derived.

### Differential gene expression analysis

Employing the 'limma' package in R, differential expression between the three CI/reperfusion injured rats and three controls in the GSE163614 dataset was conducted. DEGs were identified using a threshold of $|\log2(\text{Fold Change})| > \log2(1.2)$ and FDR $<$ 0.05. This analysis was followed by the generation of volcano and heat maps.

### Isolation of PER-related genes

PER-related pathways were sourced from the single-sample Gene-Set Enrichment Analysis (ssGSEA) official website. PER scores were computed *via* ssGSEA, and the *t*-test function was applied to compare the MCAO and sham groups. Associations between the PER score and DEGs were determined using R's 'cor' function. Genes exhibiting $|\text{cor}| > 0.7$ were designated as PER-associated DEGs. These genes were further assessed using Gene Ontology (GO) and Kyoto Encyclopedia of Genes and Genomes (KEGG) enrichment *via* the clusterProfiler package.

### Identification of central genes in PER-related genes

A protein-protein interaction (PPI) network of the PER-related DEGs was constructed using the STRING database. Essential interactions were discerned using a minimum score of 0.7. Visualization and degree analysis of the PPI were executed using Cytoscape.

### Identification of diagnostically relevant per-associated genes

For datasets GSE78731 and GSE97537, crucial genes linked to PER were scrutinized. Receiver operating characteristic (ROC) curves were produced for these DEGs, and their diagnostic efficacy was evaluated based on the Area Under the Curve (AUC).

### Correlation of key PER genes with hypoxia

To further explore the possible mechanisms of CI-reperfusion, the HIF-1 signalling pathway gene set was downloaded from the KEGG official website. HIF-1 signalling scores for MCAO and sham groups were determined using the ssGSEA approach. The correlation between key diagnostic genes and the HIF-1 pathway score was analyzed *via* the Spearman correlation method.

**Table 1  Primer sequences for 5 genes.**

| Gene | Primer name | Sequence (5′->3′) | Length | Tm | Location |
|------|-------------|-------------------|--------|------|----------|
| ADCY5 | Forward Primer | TCTCCTGCACCAACATCGTG | 20 | 62.2 | 1157-1176 |
| | Reverse Primer | CATGGCAACATGACGGGGA | 19 | 62.4 | 1326-1308 |
| CAMK2A | Forward Primer | GCTCTTCGAGGAATTGGGCAA | 21 | 62.7 | 42-62 |
| | Reverse Primer | CCTCTGAGATGCTGTCATGTAGT | 23 | 61.2 | 244-222 |
| PLCB1 | Forward Primer | GGACTGACCCTCAGGGATTTT | 21 | 61.1 | 131-151 |
| | Reverse Primer | AAGCCACGAGATTCAAATGGG | 21 | 60.6 | 364-344 |
| NTRK2 | Forward Primer | TCGTGGCATTTCCGAGATTGG | 21 | 62.7 | 155-175 |
| | Reverse Primer | TCGTCAGTTTGTTTCGGGTAAA | 22 | 60.1 | 385-364 |
| DLG4 | Forward Primer | CACAACCTCTTATTCCCAGCAC | 22 | 60.9 | 909-930 |
| | Reverse Primer | CATGGCTGTGGGGTAGTCG | 19 | 62.1 | 987-969 |
| GAPDH | Forward Primer | GGAGCGAGATCCCTCCAAAAT | 21 | 61.6 | 108-128 |
| | Reverse Primer | GGCTGTTGTCATACTTCTCATGG | 23 | 60.9 | 304-282 |

## Cell cultivation

The PC12 cell line was sourced from the Cell Bank of the Shanghai Institute of Cell Biology, affiliated with the Chinese Academy of Sciences. The cells were cultivated in Dulbecco's Modified Eagle's Medium (DMEM) supplemented with 10% fetal bovine serum (FBS; Gibco, Grand Island, NY, USA) at 37 °C. Under normoxic conditions, PC12 cells were maintained in an atmosphere consisting of 95% air and 5% $CO2$. For simulating ischemic conditions, an oxygen and glucose deprivation/reperfusion (OGD/R) model was employed. Cells were exposed to glucose-free DMEM in a hypoxic environment of 95% $N2$ and 5% $CO2$ for a duration of 2 h. Post hypoxia, cells were reintroduced to regular glucose conditions and an oxygenated environment for a recovery period of 24 h.

## qRT-PCR analysis

For assessing gene expression alterations, quantitative reverse transcription PCR (qRT-PCR) was utilized. Total RNA from PC12 cells was extracted and then reverse-transcribed using the Takara RNA PCR Kit (AMV) Ver 3.0 (TaKaRa Bio Inc., Shiga, Japan). The synthesized cDNA was subjected to PCR amplification using SYBR Green I as the fluorescent dye on an ABI PRISM 7300 real-time PCR system. The thermal cycling conditions were initiated with a denaturation step at 95 °C for 30 s, followed by 45 amplification cycles, each consisting of 95 °C for 10 s (denaturation) and 62 °C for 31 s (annealing/extension). To ensure reproducibility and accuracy, all experiments were performed in duplicate and repeated twice. The specific primer sequences used for each target gene are provided in Table 1. All primer sequences were validated by BLAST.

# RESULTS

## Screening of genes associated with CI/reperfusion injury and investigation of their relationship with PER-associated pathways

The main technical roadmap of this study is shown in Fig. S1. Differentially expressed genes (DEGs) from three rats with CI/reperfusion injury and three control rats in the

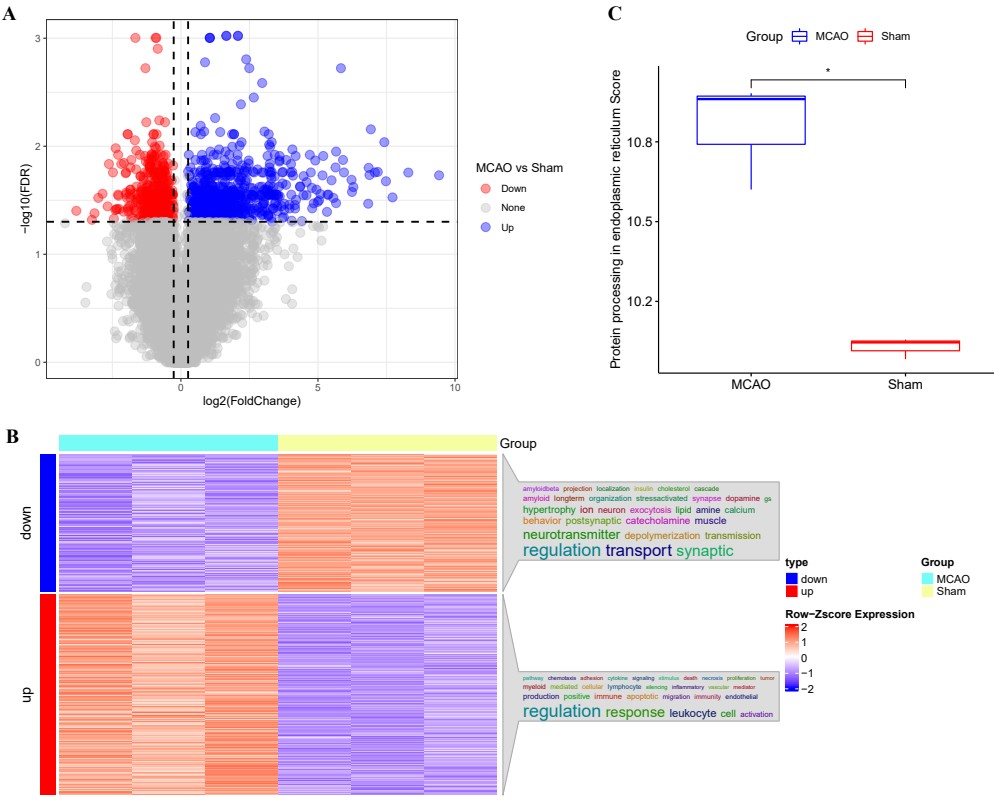

**Figure 1  Screening of genes associated with CI/reperfusion injury and their relationship with PER-associated pathways.** (A) Volcano map for the difference analysis of three rats with CI/reperfusion injury and three control rats; (B) heat map of DEGs and enrichment analysis; (C) comparison of PER-associated pathways between rats with CI/reperfusion injury and the control group (*$P < 0.05$).

GSE163614 dataset were analyzed. Both volcano (Fig. 1A) and heat maps were generated. Using the anno_GO_keywords function, an enrichment analysis word cloud (Fig. 1B) was produced, revealing 651 upregulated genes (red) and 446 downregulated genes (blue). In addition, the top 10 genes with the most significant differences in expression up- and down-regulation were summarised in Table S1, respectively. The ssGSEA analysis indicated a stronger association of CI/reperfusion injury with PER in the MCAO group, as evidenced by a higher PER score (Fig. 1C).

## Functional enrichment analysis of PER-associated gene pathways

DEGs were correlated with the PER score, and gene sets related to a |cor| > 0.7 threshold were isolated. This resulted in 648 genes positively correlated with the PER score and 446 negatively correlated genes (Table S2). Subsequent GO and KEGG enrichment analyses (using an FDR < 0.05 threshold) identified 1056 biological processes (BP), 132 cellular components (CC), 42 molecular functions (MF), and 73 KEGG signalling pathways. The top 10 enriched items are displayed as a bar graph in Fig. 2, the more the colour tends to be red, the smaller the p.adjust value, indicating a higher degree of enrichment. Importantly,

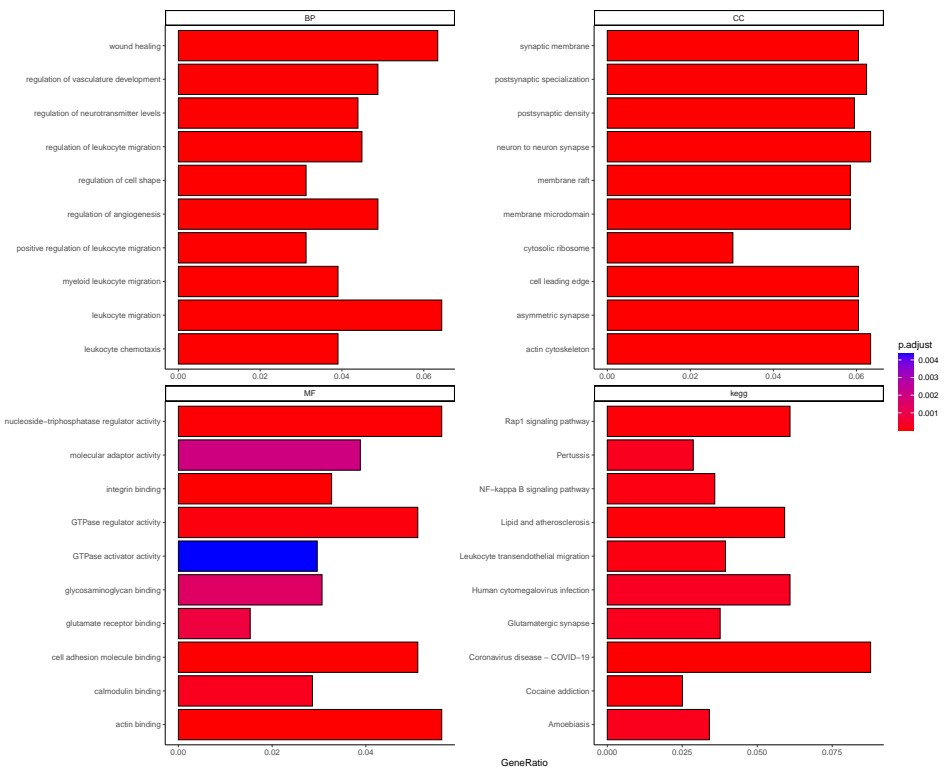

**Figure 2** **Different GO and KEGG enrichment anaysis.** Smaller adjusted p values tend towards red, indicating higher enrichment; larger adjusted p values tend towards blue, indicating lower enrichment.

the PER-associated genes showed significant involvement in neurotraumatic processes, such as leukocyte migration, wound healing, and angiogenesis regulation (Table S3).

## Identification of central genes in PER-associated gene set

The Protein-Protein Interaction (PPI) network of the PER-associated genes was built *via* the STRING database (*Mering et al., 2003*) to predict potential biological functions among proteins and further explored using Cytoscape (Fig. 3A), followed by a degree analysis (Fig. 3B). The 20 most pivotal nodes, both positively and negatively correlated, were highlighted as key genes (Table 2). A node with a higher degree value suggests its centrality within the network.

## Diagnostic potential of key PER-associated genes

For diagnostic relevance, we examined the key PER genes across the independent datasets GSE78731 and GSE97537. Seventeen of the top 20 genes showcased significant degree correlations (Figs. 4A and 4B, Table 2): ITGB1, RPS5, RPS3A, RPS27, RPL11, RPL5, RPS27L, FN1, PTPRC, RPLP0, STAT3, RPL6, RPL10A, RPL26, RPL27, RPSA, RPS15, RPS24, RPS4X, TP53, DLG4, PRKCB, GRIN2A, GNAO1, THY1, NTRK2, PRKCA, GRIA3, PLCB1, AGT, CAMK2A, DLG1, SHC3, HSPA4L, PAK1, ADCY5, APOE, NR3C1, ADRB1, RGS8. ROC curves of these 17 genes evaluated their diagnostic efficacy (Figs. 4C and 4D),

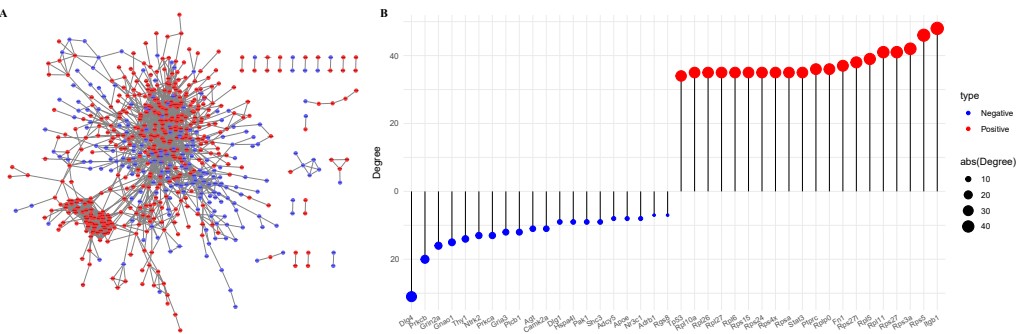

**Figure 3  Identification of key genes in PER-associated genes.** (A) PPI network of PER-associated genes, with red indicating positive correlation and blue indicating negative correlation; (B) dot-bar map of the top 20 positive and negative degree genes in the network.

**Table 2  Top 20 hub genes for relatedness.**

| Symbol | Degree | Correlation | Symbol | Degree | Correlation |
|---|---|---|---|---|---|
| *ITGB1* | 48 | Positive | *DLG4* | −31 | Negative |
| *RPS5* | 46 | Positive | *PRKCB* | −20 | Negative |
| *RPS3A* | 42 | Positive | *GRIN2A* | −16 | Negative |
| *RPS27* | 41 | Positive | *GNAO1* | −15 | Negative |
| *RPL11* | 41 | Positive | *THY1* | −14 | Negative |
| *RPL5* | 39 | Positive | *NTRK2* | −13 | Negative |
| *RPS27L* | 38 | Positive | *PRKCA* | −13 | Negative |
| *FN1* | 37 | Positive | *GRIA3* | −12 | Negative |
| *PTPRC* | 36 | Positive | *PLCB1* | −12 | Negative |
| *RPLP0* | 36 | Positive | *AGT* | −11 | Negative |
| *STAT3* | 35 | Positive | *CAMK2A* | −11 | Negative |
| *RPL6* | 35 | Positive | *DLG1* | −9 | Negative |
| *RPL10A* | 35 | Positive | *SHC3* | −9 | Negative |
| *RPL26* | 35 | Positive | *HSPA4L* | −9 | Negative |
| *RPL27* | 35 | Positive | *PAK1* | −9 | Negative |
| *RPSA* | 35 | Positive | *ADCY5* | −8 | Negative |
| *RPS15* | 35 | Positive | *APOE* | −8 | Negative |
| *RPS24* | 35 | Positive | *NR3C1* | −8 | Negative |
| *RPS4X* | 35 | Positive | *ADRB1* | −7 | Negative |
| *TP53* | 34 | Positive | *RGS8* | −7 | Negative |

with each gene demonstrating AUCs exceeding 0.7, suggesting their diagnostic promise. Further validation in the GSE163614 dataset supported these findings (Fig. 5).

## Evaluating the association of diagnostic key PER genes with hypoxia

Decreased cerebral blood flow can instigate hypoxia, precipitating ischemic neuronal injury and favouring anaerobic tissue metabolism (*Kalogeris et al., 2012*). To contextualize hypoxia, the HIF-1 signalling pathway was extracted from KEGG. Its scores across groups

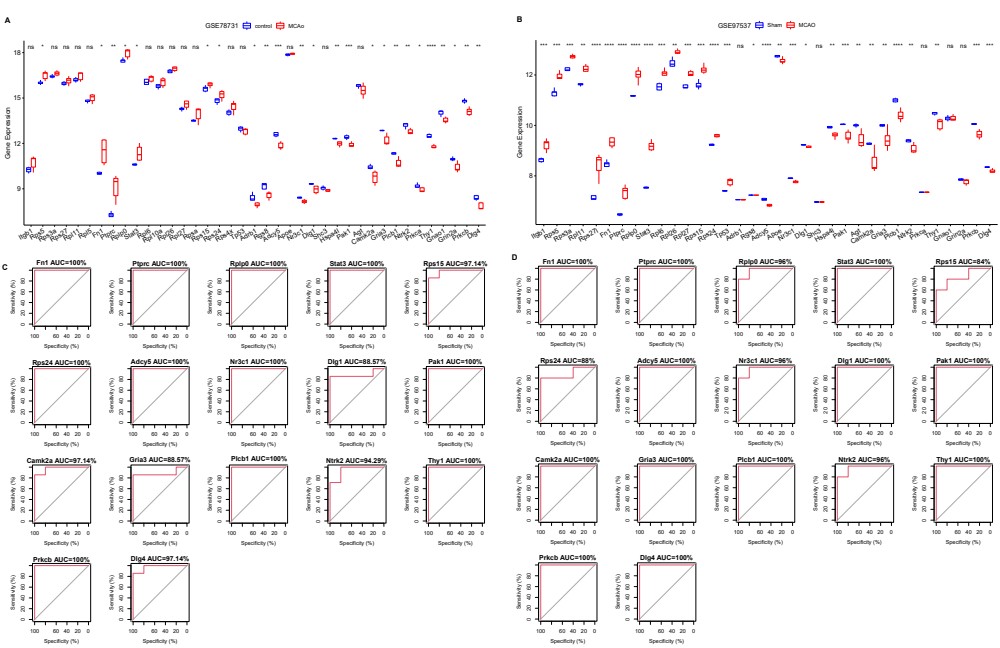

**Figure 4  Screening of key genes with diagnostic value.** Comparison of expression levels of the top 20 degree genes with positive/negative correlations in GSE78731 (A) and GSE97537 (B); ROC curves in GSE78731 (C) and GSE97537 (D). ***$P < 0.05$; ****$P < 0.01$; *****$P < 0.001$.

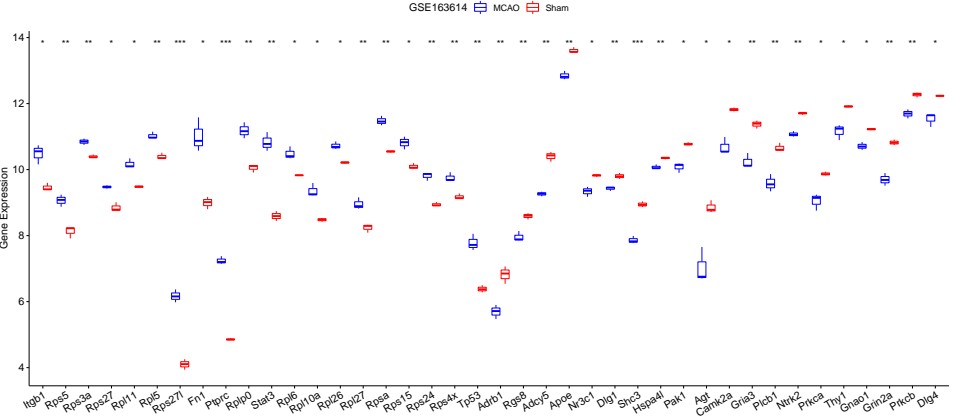

**Figure 5  Notable differences were found among the top 20 degree genes with positive/negative correlation between the MCAO and sham groups in GSE163614.** *$P < 0.05$; **$P < 0.01$; ***$P < 0.001$.

were computed using ssGSEA, revealing an elevated score in the MCAO group (Fig. 6A). Correlation analyses between the 17 diagnostic genes and the HIF-1 signalling pathway score identified six genes—PRKCB, ADCY5, CAMK2, PLCB1, NTRK2, and DLG4—that displayed significant associations with hypoxia (Fig. 6B).

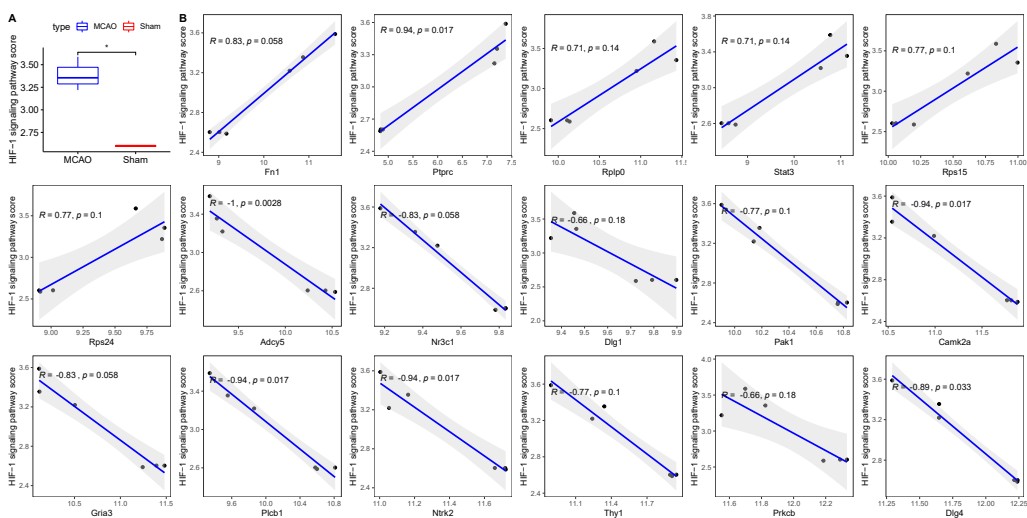

**Figure 6** **Relationships between 17 key PER genes with diagnostic value and hypoxia.** (A) HIF-1 signaling pathway score in rats with CI/reperfusion injury and control group; (B) correlation analysis between 17 key genes and HIF-1 signaling pathway score.

## Investigating the interplay between key hypoxia-linked genes and apoptosis

Exacerbated endoplasmic reticulum stress (ERS) activation during CI/reperfusion can stimulate apoptosis *via* pathways such as CHOP, Caspase12, and JNK (*Wang et al., 2022*). We probed the connections between crucial hypoxia-associated genes and apoptosis. The MCAO group displayed a marked under-expression of these genes compared to the Sham group (Fig. 7). Subsequent analyses showed an elevated apoptosis score in the MCAO group (Fig. 8A). Furthermore, ADCY5, CAMK2A, PLCB1, NTRK2, and DLG4 showed significant negative correlations with apoptosis score (Fig. 8B). These genes appear to be instrumental in hypoxia-induced apoptosis during CI/reperfusion injury. PCR-based validation further highlighted their reduced expression in cells subjected to glucose deprivation/reperfusion (OGD/R), which suggests their consistent downregulation during such metabolic stress (Fig. 9).

## DISCUSSION

CI is a leading cause of disability and mortality worldwide (*Siniscalchi et al., 2014*). Typically initiated by an embolism or thrombosis in a major cerebral artery (*Miller et al., 2017*), CI is primarily managed through thrombolysis and restoration of blood flow (*Jean et al., 1998*). Although acute ischemia inflicts neuronal damage, the subsequent restoration of blood flow can intensify this injury, culminating in significant brain dysfunction (*Rosenberg, Estrada & Dencoff, 1998*). The challenge, therefore, is to prevent the amplification of brain injury following ischemia/reperfusion.

In the current study, the ssGSEA algorithm was used for the first time to analyse the PER scores of MACO rats. The results showed that the PER score was increased in the MCAO

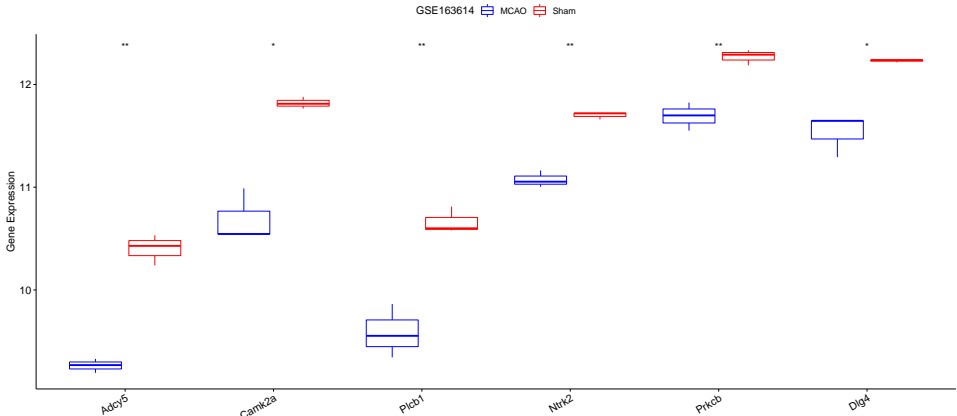

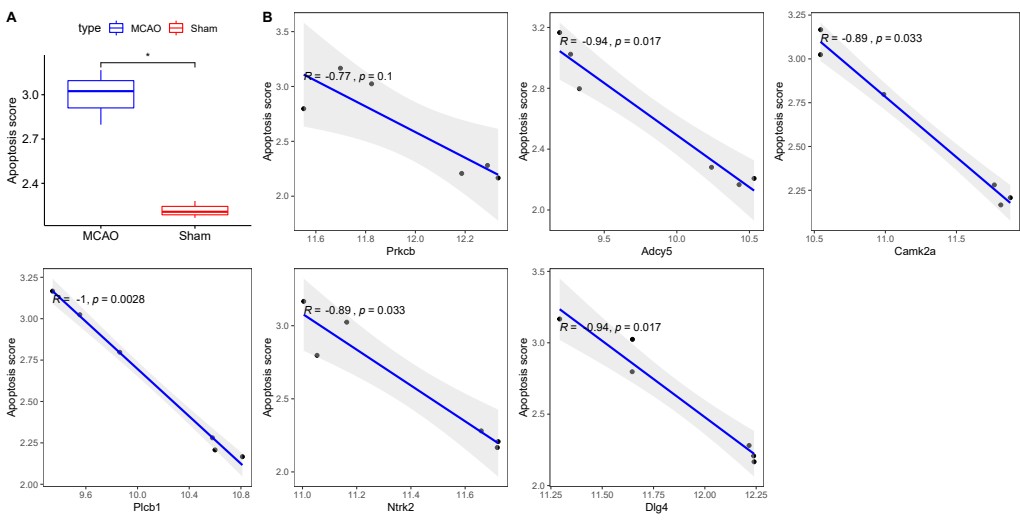

**Figure 7 Expression analysis of six key hypoxia-associated genes in the GSE163614 dataset.** ***P < 0.05; ****P < 0.01.

**Figure 8 Correlations between key hypoxia-associated genes and apoptosis.** (A) Comparison of apoptosis score between rats with CI/reperfusion injury and control group; (B) correlation analysis between six key hypoxia-associated genes and apoptosis score.

group, which suggests a close association between MCAO and PER. Subsequent DEG and correlation analyses were employed to identify genes with a strong PER affiliation, leading to the construction of a PPI. Within this PPI, 40 core genes were discerned. Utilizing two external validation datasets, we affirmed the expression and diagnostic relevance of these core nodes for the MCAO group, pinpointing 17 essential diagnostic genes. Considering the established link between ischemia/reperfusion-induced apoptosis and hypoxia, we further discerned five genes (ADCY5, CAMK2A, PLCB1, NTRK2, and DLG4) from the initial 17 that displayed significant associations with both hypoxia and apoptosis. These

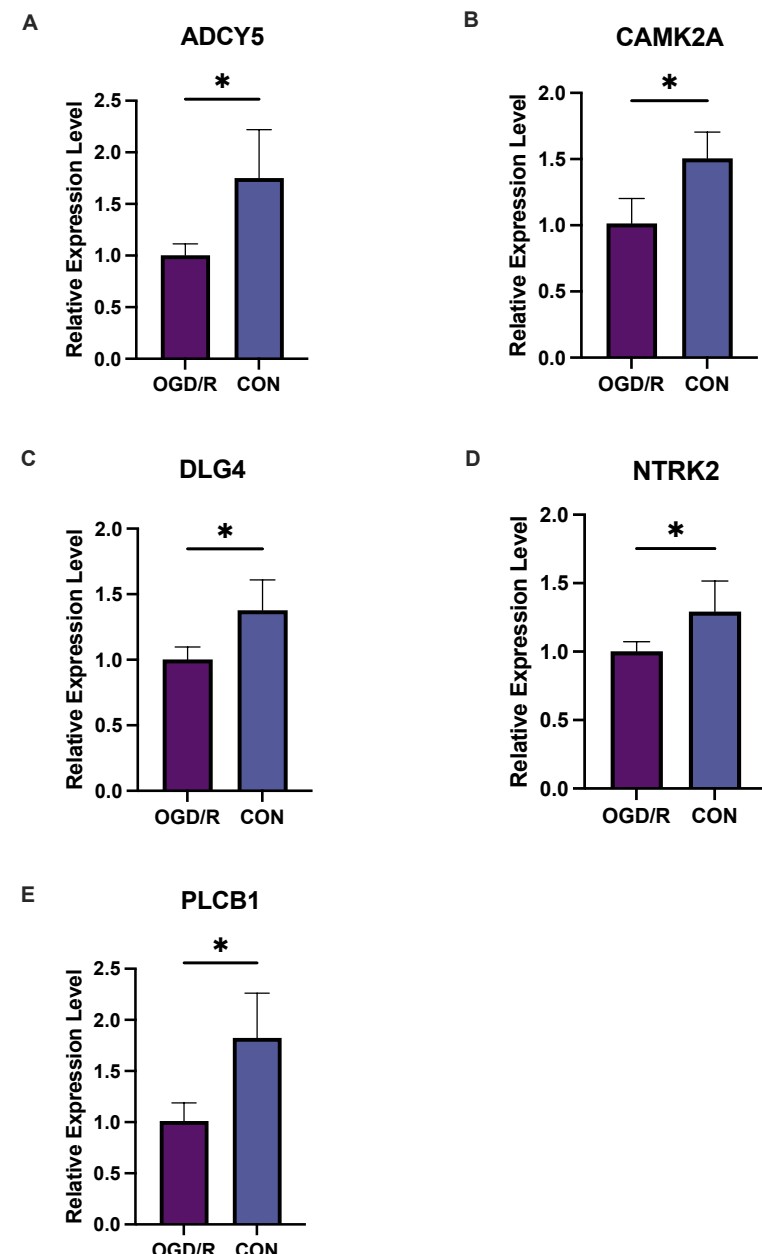

**Figure 9** **Five key genes show low expression in PC12 cells from an oligosaccharide/reperfusion (OGD/R) injury model.** The relative expression levels of (A) ADCY5, (B) CAMK2A, (C) DLG4, (D) PLCB1, and (E) NTRK2 in PC cells were examined using PCR. *$P < 0.05$; ***$P < 0.001$.

genes are hypothesized to be paramount in instigating apoptosis in hypoxic conditions during ischemia/reperfusion injury.

ADCY5, an adenylate cyclase family member, facilitates the conversion of adenine-5′-triphosphate to 3′, 5′-cyclic adenosine-5′-triphosphate (cAMP) (*Linder, 2006*). Though ADCY5 is ubiquitously present in the brain and has been linked to various neuro-related

conditions (*Ferrini et al., 2021*; *Kaiyrzhanov et al., 2021*), ADCY5 is specifically expressed in the brain and selectively possesses high expression levels in sites such as the striatum and olfactory nodes, and mutations in its gene may affect neurodevelopmental phenotypes, which in turn may lead to motor dysfunction. Its association with CI/reperfusion remains unexplored. We conjecture that ADCY5 may also influence the quality of survival of survivors by modulating motor function in CI/reperfusion.

CAMK2, a calcium-responsive serine/threonine kinase (*Chao et al., 2011*), comprises four distinct subunits (CAMK2A, 2B, 2D, and 2G) each possessing specific tissue distributions and functionalities (*Wang et al., 2020a*). Especially noteworthy are the CAMK2A and CAMK2B variants, which dominate brain expression (*Cook et al., 2018*; *Wang et al., 2020a*). Not only is CAMK2A implicated in cognitive functions (*Lee et al., 2021*), neurodevelopment (*Akita et al., 2018*), and certain cancer types (*Wang et al., 2020a*; *Yu et al., 2021*), but a study parallel to ours also unveiled its potential to augment synaptic plasticity in MCAO mice (*Shen et al., 2021*). In addition, it has been shown that CAMK2A may confer neuroprotection to neurons in CI/reperfusion through NF-KB signalling (*Ye et al., 2019*).PLCB1, primarily localized in cerebral regions (*Caricasole et al., 2000*; *McOmish et al., 2008*), has been associated with various neuropsychiatric disorders (*Girirajan et al., 2013*; *Schoonjans et al., 2016*; *St Pourcain et al., 2014*; *Udawela et al., 2011*; *Vasco, Cardinale & Polonia, 2012*). The NTRK2 gene, encoding the tropomyosin receptor kinase B (TrkB), plays an instrumental role in neuronal development and survival (*Amatu, Sartore-Bianchi & Siena, 2016*), findings which resonate with our study. This alludes to the potential therapeutic benefits of augmenting TRKB expression to attenuate ischemia/reperfusion injury. It has been shown that PLCB1, a key biological marker in cognitive improvement, modulates PLCB1 expression abundance in the cerebral cortex and contributes to the improvement of associated cognitive dysfunction (*Mabondzo et al., 2023*).

DLG4 is responsible for synthesizing the post-synaptic density protein 95 (PSD-95), a member of the membrane-associated guanylate kinases family. As a predominant scaffold protein in excitatory postsynaptic density, it is instrumental in synaptic plasticity (*Bustos et al., 2017*; *Rodríguez-Palmero et al., 2021*). Its aberrant expression has been observed in numerous neurological disorders (*Arbuckle et al., 2010*; *Bustos et al., 2017*; *De Bartolomeis et al., 2014*; *Savioz, Leuba & Vallet, 2014*; *Zhang et al., 2014*), with previous research suggesting that enhancing PSD95 expression may bolster synapse count and minimize neuronal death (*Wang et al., 2020b*). Previous studies have shown that the use of electrical stimulation of the parietal nucleus of the cerebellum can modulate neurotransmitter release from synaptic vesicles by methylating DLG4 and activating neuronal and neurovascular coupling, thereby stimulating interneuronal neuroprotection and reducing brain injury (*Gao et al., 2023*).

Our study delineates five key genes (ADCY5, CAMK2A, PLCB1, NTRK2, and DLG4) that may be instrumental in prompting apoptosis in hypoxic scenarios during CI/reperfusion injury. While some genes' roles in mitigating CI/reperfusion injury have been established, others require further validation. Nonetheless, their extensive brain expression and crucial functions are indisputable. It is imperative to further investigate the notion that

modulation of these genes' expression can influence cell apoptosis *via* PER, culminating in CI/reperfusion injury.

### Funding
This work was supported by the Science and Technology Project: Yinzhou District Ningbo City (2021AS0054). The funders had no role in study design, data collection and analysis, decision to publish, or preparation of the manuscript.

### Grant Disclosures
The following grant information was disclosed by the authors:
Science and Technology Project: Yinzhou District Ningbo City: 2021AS0054.

### Competing Interests
The authors declare there are no competing interests.

### Author Contributions
- Liang-da Li conceived and designed the experiments, analyzed the data, prepared figures and/or tables, and approved the final draft.
- Yue Zhou performed the experiments, authored or reviewed drafts of the article, and approved the final draft.
- Shan-fen Shi analyzed the data, prepared figures and/or tables, and approved the final draft.

### Data Availability
The raw data is available at figshare: Li, Liangda (2023). Raw data.zip. figshare. Journal contribution. https://doi.org/10.6084/m9.figshare.22632745.v1.

### Supplemental Information
Supplemental information for this article can be found online at http://dx.doi.org/10.7717/peerj.16707#supplemental-information.

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
