# Peer review of "Identification and characterization of biomarkers associated with endoplasmic reticulum protein processing in cerebral ischemia-reperfusion injury"

_PeerJ, doi:10.7717/peerj.16707_

## Round 0.1 · original submission · Major Revisions

Authors should mention and discuss the novelty of this study. There have been several studies that focused on cerebral ischaemia-reperfusion injury.

Please provide more details about PCR validation, which is mentioned in the text.

It is better to invite a fluent English speaker to edit the language.

**Language Note:** The Academic Editor has identified that the English language must be improved. PeerJ can provide language editing services - please contact us at [email protected] for pricing (be sure to provide your manuscript number and title). Alternatively, you should make your own arrangements to improve the language quality and provide details in your response letter. – PeerJ Staff
Please make full revisions according to reviewers' comments.

Reviewer 1 ·

Basic reporting

1. fair use of language throughout
2. publicly available datasets have been used in this study and it is recommended that authors provide detailed citations indicating the source of the original data so that the reader can further understand the context and background of the data source.
3. graphs are mentioned in several places (e.g. Fig. 1A, Fig. 2, etc.), but during the review process I would have preferred greater clarity in all graphs, as well as accuracy and completeness in the notes and captions.

Experimental design

1. more literature is needed in the introduction describing how the study fills identified gaps in knowledge
2. PCR validation was mentioned in the study but the primer sequences used were not explicitly reported. Given the importance of reproducibility, it is recommended that full information on primer sequences is provided, including whether these primers have been validated.

Validity of the findings

1. a relatively large number of similar studies have been performed in cerebral ischaemia-reperfusion injury, which is the novelty of the current study.
2. In the Results section, there are several references to various statistical tests, but the parameters or values of the tests are not clearly stated. To ensure the accuracy and reproducibility of the statistical results, it is recommended that more detailed information on the statistical tests, such as p-values, confidence intervals, etc., be provided.
3. the conclusions are well formulated, relevant to the original research question and limited to supporting the results, but it is recommended to streamline the language description.

Additional comments

For ADCY5, CAMK2A, PLCB1, NTRK2 and DLG4, which are designated as key genes, it is suggested to further discuss their expression and function in other related diseases or disease models. This will help readers to better understand the specificity and importance of these genes in cerebral ischaemia-reperfusion injury.

Reviewer 2 ·

Basic reporting

The article "Identification and Characterisation of Biomarkers Associated with Endoplasmic Reticulum Protein The article "Identification and Characterisation of Biomarkers Associated with Endoplasmic Reticulum Protein Processing in Cerebral Ischemia-Reperfusion Injury" was written with appropriate methodology and clear logical structure. However, there are some parts that could be improved.
1. In the abstract, the authors have successfully highlighted the possible roles of five key genes (ADCY5, CAMK2A, PLCB1, NTRK2 and DLG4) in cerebral ischaemia-reperfusion injury. However, a more detailed explanation of the importance of these genes in previous studies is suggested in the introduction to provide the reader with more background knowledge.
2. In the Results section, although extensive data and graphs are provided to support the authors' conclusions, it is suggested that further explanation of how the five genes were determined to be associated with hypoxia and apoptosis be provided in the subsection "Interaction studies on key genes associated with hypoxia".
3. In the Discussion section, the authors have explained in detail the possible functions and importance of the five key genes mentioned. However, whether some of these five genes are more important in terms of their expression and function under hypoxic conditions requires further discussion.
4. In the Methods section, the bioinformatics methods and techniques used are described in detail. However, in the subsections "Single cell culture" and "qRT-PCR analysis", it is recommended to provide more details, such as the treatment time of the cells and the specific experimental conditions, in addition to the section on apoptosis, which is relatively redundant and should be excluded.
5. Overall, this is a thorough research article that provides new insights into five key genes involved in cerebral ischaemia-reperfusion injury. However, to make the article more convincing, further experimental validation is recommended, such as manipulating the expression of these genes in model animals to observe their effects on ischaemia-reperfusion injury.

Experimental design

None

Validity of the findings

None

Additional comments

None

Reviewer 3 ·

Basic reporting

language would hopefully be improved on a large scale
although the abstract is generally well written, it is recommended that the authors provide more background on the function of the five key genes and how these five key genes specifically relate to cerebral ischaemia-reperfusion injury.
it is recommended that more information on the mechanisms of cerebral ischaemia-reperfusion injury be included in the introductory section to give the reader a more comprehensive background.
A graphical summary is desirable to summarise the findings of this study.

Experimental design

Three different data sets are described in section 2.1, but the role of each data set is not made clear. It is recommended that more justification is given for the selection and use of each dataset. In subsection 2.6, it is recommended that more detailed information is provided on how the HIF-1 pathway gene set was selected.

Validity of the findings

In subsection 3.1 "Screening of genes associated with CI/reperfusion injury", the authors mention 651 up-regulated genes and 446 down-regulated genes, but do not list the specific names of these genes. To allow the reader to better follow the progress of the study, it is recommended that a complete list of these genes be provided or included in the Supplementary Material. In Section 3.3, the PPI network constructed using the STRING database may be confusing to readers who are not familiar with this database. It is suggested that the authors add a short description explaining the rationale for the choice of this database and how it can help the research.
It is recommended that more external studies are included in the discussion section to compare with your findings, which will give the reader a broader context. When discussing the function of the five key genes, it is recommended that the authors explain more specifically how these genes interact with cerebral ischaemia-reperfusion injury. It is recommended that the authors more clearly summarise the key findings of this study and briefly discuss their potential clinical applications in the conclusion section.

Additional comments

None.

---

## Round 0.2 · Minor Revisions

Authors should thoroughly check the paper and make several revisions.
In line 183-184, what's the mean of the "[10]"? Maybe it is a citation, please correct it.
Please edit to place a space prior to all parentheses
More description and explantation of data should be put in each results section and all figure legends.
For example, in 3.1, "651 upregulated genes (red) and 446 downregulated genes (blue)" - what are these, maybe talk about the most significantly different, or most highly down and up regulated.
In the discussion, authors stated that " Utilizing two external validation datasets, we affirmed the expression and diagnostic relevance of these core nodes for the MCAO group, pinpointing 17 essential diagnostic genes." Please describe 17 essential diagnostic genes in the results.
"Considering the established link between ischemia/reperfusion-induced apoptosis and hypoxia, we further discerned five genes (ADCY5, CAMK2A, PLCB1, NTRK2, and DLG4) from the initial 17 that displayed significant associations with both hypoxia and apoptosis." Please add related results in the result section.
The mentioned cutoff value ("threshold of |log2(Fold Change)| > log2(1.2)") may be too lenient, usually more meaningful results are obtained with > log2(2)? There have been so many DEGs, can authors get a more meaningful analysis with a higher fold change?
In Fig. 2, what do the colors mean? The resolution and size of labels is hard to read, Authors should decribe more in legends.
Please move Table 2 to supplementary files.
Table 3 says "Top 10 entries . . . ", but there are more than 10 entries? Need to make sure that the columns are justified, and if not, this also may be better as a supplemental excel file.

Reviewer 1 ·

Basic reporting

These areas have been significantly modified.

Experimental design

no comment.

Validity of the findings

No more problems.

Additional comments

I recognise the author's revision.

Reviewer 2 ·

Basic reporting

The manuscript is accepted for publication in its current version.

Experimental design

The manuscript is accepted for publication in its current version.

Validity of the findings

The manuscript is accepted for publication in its current version.

Additional comments

None

Reviewer 3 ·

Basic reporting

no comment

Experimental design

no comment

Validity of the findings

no comment

Additional comments

The author's revisions have resolved my earlier confusion

---

## Round 0.3 · accepted · Accept

Authors have made corrections according to my comments. This paper can be accepted.